# Non-Invasive Neuromodulation Methods to Alleviate Symptoms of Huntington’s Disease: A Systematic Review of the Literature

**DOI:** 10.3390/jcm12052002

**Published:** 2023-03-02

**Authors:** Lijin Jose, Lais Bhering Martins, Thiago M. Cordeiro, Keya Lee, Alexandre Paim Diaz, Hyochol Ahn, Antonio L. Teixeira

**Affiliations:** 1Neuropsychiatry Program, Department of Psychiatry and Behavioral Sciences, The University of Texas Health Science Center, Houston, TX 77054, USA; 2Texas Medical Center Library, Houston, TX 77030, USA; 3Center for the Study and Prevention of Suicide, Department of Psychiatry, University of Rochester Medical Center, Rochester, NY 14642, USA; 4College of Nursing, Florida State University, Tallahassee, FL 32306, USA

**Keywords:** Huntington’s disease, neuromodulation, ECT, TMS, tDCS, neuropsychiatric symptoms

## Abstract

Huntington’s disease (HD) is a progressive and debilitating neurodegenerative disease. There is growing evidence for non-invasive neuromodulation tools as therapeutic strategies in neurodegenerative diseases. This systematic review aims to investigate the effectiveness of noninvasive neuromodulation in HD-associated motor, cognitive, and behavioral symptoms. A comprehensive literature search was conducted in Ovid MEDLINE, Cochrane Central Register of Clinical Trials, Embase, and PsycINFO from inception to 13 July 2021. Case reports, case series, and clinical trials were included while screening/diagnostic tests involving non-invasive neuromodulation, review papers, experimental studies on animal models, other systematic reviews, and meta-analyses were excluded. We have identified 19 studies in the literature investigating the use of ECT, TMS, and tDCS in the treatment of HD. Quality assessments were performed using Joanna Briggs Institute’s (JBI’s) critical appraisal tools. Eighteen studies showed improvement of HD symptoms, but the results were very heterogeneous considering different intervention techniques and protocols, and domains of symptoms. The most noticeable improvement involved depression and psychosis after ECT protocols. The impact on cognitive and motor symptoms is more controversial. Further investigations are required to determine the therapeutic role of distinct neuromodulation techniques for HD-related symptoms.

## 1. Introduction

Huntington’s disease (HD) is an autosomal dominant neurodegenerative disease characterized by progressive motor, cognitive, and behavioral symptoms [1,2] The expansion of glutamine (CAG) repeats in the coding region of the *Huntingtin* gene leads to the mutated protein [3]. The average age at onset of signs and symptoms is 40 years old, with death occurring within 15–20 years later [4].

Currently, there are no disease modifying treatments for HD, but several pharmacological approaches have been proposed to manage HD-related symptoms. Vesicular monoamine transporter 2 (VMAT2) inhibitors (e.g., deutetrabenazine) are effective against chorea, but can cause somnolence and weight gain [5,6]. Psychotropic medications (e.g., antidepressants, anti-psychotics) are used for behavioral management, but they are also associated with a plethora of adverse effects, including motor symptoms, weight gain, sexual dysfunction [7]. Non-pharmacological approaches, including different modalities of psychosocial intervention, have been used for the management of patients with HD as well.

The use of neuromodulation techniques for the management of HD-related symptoms is not FDA approved, and their use is still restricted to the context of research. Nonetheless, there has been a growing interest in the potential therapeutic role played by non-invasive neuromodulation methods, such as electroconvulsive therapy (ECT), transcranial magnetic stimulation (TMS), and transcranial electric stimulation (tES), especially transcranial direct current stimulation (tDCS), in neurodegenerative diseases, including HD [8,9].

The neuropathological hallmarks of HD include loss of GABAergic neurons and atrophy of the striatum [10]. As the disease progresses, degeneration spreads to other brain regions. The pathophysiology of the disease is also characterized by functional neural changes that might even precede structural alterations [11]. A recent systematic review showed that patients with HD display aberrant brain connectivity in the sensory, motor, visual, and executive/attentional networks [12]. The mechanisms of action of non-invasive neuromodulation methods are believed to involve the reorganization of brain networks [13,14,15]. Therefore, non-invasive neuromodulation methods might have the potential to change the aberrant connectivity seen in HD, positively influencing the related symptoms. Indeed, preliminary studies suggest that non-invasive neuromodulation methods can be used in the treatment of HD-related symptoms with minimal or manageable side effects and maximum effectiveness, but the evidence is still sparse. In this context, we performed a systematic review of studies assessing the effects of different non-invasive neuromodulation methods such as ECT, TMS, and tDCS on HD-related motor, cognitive, and behavioral symptoms.

## 2. Methods

This study was registered at (PROSPERO 2021 CRD42021255823) and can be accessed at https://www.crd.york.ac.uk/prospero/display_record.php?ID=CRD42021255823. The information sources used were Ovid MEDLINE (1946–13 July 2021), Cochrane Central Register of Clinical Trails (July 2021), Embase (1974–13 July 2021), and PsycINFO (1806–13 July 2021). The search was limited to the English language. Search terms were included in the Appendix A.

### 2.1. Study Selection

After duplicates had been removed, two independent researchers reviewed all references found through database searching. If needed, a third independent reviewer was called upon to resolve conflicts. Records were screened and excluded based on eligibility criteria (see Appendix A). Briefly, the eligibility criteria included studies that investigated the effects of non-invasive neuromodulation on HD-related symptoms by comparing stimulation vs. sham or standard treatment. Only studies written in the English language were considered, and studies were excluded if they used any modality of invasive neuromodulation (e.g., DBS, VNS) or peripheral nerve stimulation. Other exclusion criteria included screening and diagnostic tests, such as electrophysiological studies, experimental studies on animal models, reviews and systematic reviews, and meta-analysis.

### 2.2. Data Collection and Data Items

Data were collected by an independent researcher and included the following: study type, sample size, intervention (ECT: stimulation parameter, seizure duration, electrode position, total sessions; TMS: stimulation parameters, total duration, frequency, electrode position, interval period; tDCS: anodal vs. sham stimulation parameters, total duration, frequency, electrode position, interval period), tools (motor, psychopathological and cognitive measures), outcome (primary, secondary), and side effects.

### 2.3. Quality Assessment

Two independent researchers assessed the quality of each study using the Joanna Briggs Institute’s (JBI) critical appraisal tools. A third researcher resolved discrepancies. The JBI critical appraisal checklist contains 8 items for case-control studies, 10 items for case-series studies, and 13 items for RCTs. Each item is categorized as ‘yes’, ‘no’, ‘unclear’, or ‘not applicable’ (N/A). Each study was scored based on whether it met the inclusion criteria. One point was assigned to all fields with a ‘yes’ answer choice and 0 points were allotted to all other answer choices (‘no’, ‘unclear’, N/A). Higher scores indicate better quality [16].

## 3. Results

### 3.1. Study Selection

A total of 750 studies were initially retrieved, and after duplicates were removed, 611 abstracts were screened. Of those, 574 were excluded based on lack of fulfillment of inclusion and exclusion criteria. The most common reason for exclusion was the use of neuromodulation, not as a therapeutic but as an investigative tool for HD. Ultimately, 37 full-text articles were assessed for eligibility, of which 18 were excluded, leaving 19 articles that were included in the qualitative synthesis. Figure 1 shows the details of the study selection and exclusion reasons.

### 3.2. Study Characteristics

The characteristics of the studies included in this systematic review are displayed in Table 1, Table 2 and Table 3. Thirteen studies used ECT (ten case reports and three case series), four TMS (two RCTs, one case report, and one case series), and two tDCS (both RCTs) in the treatment of HD. No study using other neuromodulation methods such as transcranial alternating current stimulation (tACS), transcranial pulsed current stimulation (tPCS), or transcranial random noise stimulation (tRNS) were found in HD. The average score for quality assessment was 5/8 on case-report studies, 6/10 on case-series studies, and 8.5/13 on RCTs. Therefore, their overall quality was moderate (see Appendix A). 

## 4. ECT

### 4.1. Study Design

We identified 13 studies that used ECT to treat HD symptoms: 10 case reports [17,19,20,21,22,24,25,26,28,29], and 3 case series [18,23,27] (Table 1). A total of 27 patients with HD were enrolled (13 female and 14 male participants) in these studies, with an age range between 26 and 65. The mean number of ECT sessions was 12.7 (standard deviation [SD] = 9.13). Maintenance ECT ranged from one session every one week to every four weeks. Assessments were made at baseline for all studies. Post-intervention assessments were described in seven studies.

### 4.2. Intervention Parameters

Five out of thirteen studies did not report intervention parameters. Stimulation parameters varied significantly among studies that reported them. Five studies did not report any information about the parameters used for ECT [17,22,25,26,28]. Two studies described the method used to set the parameters but did not describe specific parameters [21,24]. Within the six studies that described any specific parameters, frequency varied between 10 and 140 Hz, pulse range between 1.0 and 1.6 ms, and dynamic energy between 33.3 and 55.7 Joules. Electrode position was not reported in three out of thirteen studies. In the remaining ten studies, right unilateral (RUL) placement was used in three studies, bilateral (BL) in six studies, and one study reported the use of both RUL and BL.

### 4.3. Assessment Tools and Outcomes

Motor symptoms were evaluated in three studies using the Unified Huntington’s Disease Rating Scale (UHDRS) [25,28]. Three studies reported Total Functional Capacity (TFC) scores [19,25,28]. Cognition was evaluated through the Mini-Mental State Exam (MMSE) in four studies [18,19,22,25] and the Montreal Cognitive Assessment in two studies [27,28]. Different clinical tools were used in psychiatric assessment. Behavioral symptoms were assessed using Hamilton Depression Rating Scale (HAM-D) [19], Bush-Francis Catatonia Rating Scale (BFCRS) [21], Positive and Negative Syndrome Scale (PANSS) [22], Brief Psychiatric Rating Scale (BPRS) [22,25], Clinical Global Impression (CGI) [25], and Montgomery–Åsberg Depression Rating Scale (MADRS) [25,28]. Neuroimaging was performed at baseline and post-intervention in two studies: one employed computed tomography (CT) scan [26] and the other single-photon emission computed tomography (SPECT) scan [22].

Twelve of the thirteen ECT studies reported improvement in different behavioral symptoms, such as psychosis, depression, irritability, agitation, suicidal and homicidal tendencies [17,19,20,21,22,23,24,25,26,27,28,29]. The most frequent behavioral symptoms that improved with ECT were depressive symptoms—6 out of 13 studies [17,19,23,25,27,28]—and psychosis—7 out of 13 studies [17,18,22,23,24,27,29]. Seven [18,20,23,24,25,26,27] and six studies [18,19,23,24,27,28] also found improvements in motor and cognitive symptoms, respectively. Two ECT studies [18,23] reported new or worsening behavioral symptoms. Cusin et al. [23] reported two cases of transient agitation after anesthesia. Ranen et al. [18] reported a case of worsening catatonia with concomitant cognitive improvement, and a case of new psychotic symptoms and cognitive impairment. Two studies reported transient cognitive impairment that reversed after discontinuing ECT [22,23]. One study [18] reported a case of non-transient cognitive deterioration. Three out of the twenty-seven patients from the thirteen ECT studies presented worsening of motor symptoms [17,18,28].

## 5. TMS

### 5.1. Study Design

We identified four studies that used TMS for HD: two crossover trials [30,31], one case series [32], and one case report [33] (Table 2). A total of 15 patients with HD were enrolled (four female, five male participants, six participants whose gender was not mentioned) in these studies, with an age range between 32 and 77. One crossover trial was randomized and blinded [31], but the second study was pseudorandomized [30]. The number of sessions ranged between one and forty-nine sessions. Assessments were performed at baseline and post-intervention in all studies. The latter assessments varied significantly from minutes [30,31,32] up to days or weeks post-intervention.

### 5.2. Intervention Parameters

Three studies used repetitive TMS (rTMS) [30,31,32] and one study used deep repetitive TMS (dTMS) [33]. Three studies used figure-of-eight coils [30,31,32] and one used an H coil [35]. All rTMS studies stimulated at an intensity of 90% motor threshold, with stimulation frequency between 1 Hz and 10 Hz. The number of rTMS sessions ranged from one to seven. In two studies, the stimulation focused on the supplementary motor area of both hemispheres [30,32], while one stimulated the left primary motor cortex [31]. All studies used head surface anatomical landmarks to define stimulation targets. Electromyography was used to determine motor threshold in two studies [30,31,32], while two studies [30,32,33] did not report the method used to define the motor threshold (i.e., electromyography, observation of muscle twitch). In the rTMS study, stimulation was performed over the right dorsolateral pre-frontal cortex at an intensity of 120% motor threshold, frequency of 1 Hz, delivering 1600 pulses for 49 daily sessions [33].

### 5.3. Assessment Tools and Outcomes

Motor symptoms were evaluated with the Unified Huntington’s Disease Rating Scale (UHDRS) [31,33], Abnormal Involuntary Movement scale (AIMs) [30,32], and Nine Hole Peg Test (NHPT) [31]. Behavioral symptoms were assessed with Beck Depression Inventory (BDI) [31] and Geriatric Depression Scale (GDS) [33].

One study reported improvement of depressive symptoms after dTMS [33], while another described improvement of depressive symptoms and mixed effects over motor symptoms after rTMS [31]. One study reported transient improvement of motor symptoms after rTMS [30]. One study did not report any benefit from rTMS [32]. Of the fifteen patients included in the TMS studies, only two presented side effects, such as transient worsening of bradykinesia, eye lacrimation, and discomfort at the site of stimulation.

## 6. tDCS

### 6.1. Study Design

There were two double-blinded crossover trials with tDCS [34,35] involving 24 participants with age ranging from 43 to 72 years (Table 3). The interval between the active and sham stimulations varied from one week to three months. In one study, there were only two sessions (one sham and one anodal stimulation) [34], and there were a total of five sessions in the second study [35]. Assessments were performed at baseline and at the end of treatment in both studies, while one also evaluated the patients four weeks after the end of the trial [35].

### 6.2. Intervention Parameters

One study employed tDCS stimulation with the anode placed on the frontal area (F3 in 10–20 EEG system) to stimulate the left dorsolateral pre-frontal cortex (DLPFC), while the cathode was placed on the contralateral orbital area (FP2 in 10–20 EEG system) [34]. The other study stimulated the cerebellum (ctDCS) with the anode placed on the median line, 2 cm below the inion, with lateral borders about 1 cm medially to the mastoid apophysis, and the cathode over the right shoulder [35]. Both studies had around 20 min sessions, with a current intensity of 1.75 mA.

### 6.3. Assessment Tools and Outcomes

Motor symptoms were evaluated using the UHDRS [34,35]. Cognitive assessments were performed with the Wechsler Adult Intelligence Scale (WAIS), Digit ordering test-adapted (DOT-A), 1-back, 2-back (modified version of the n-back task), and stroop task [34]. Behavioral symptoms were evaluated using the Hospital Anxiety and Depression scale (HADS) [34].

Both trials resulted in positive effects in motor [35] and cognitive [34] symptoms. Out of twenty-four participants from both studies, six reported tingling, five reported itching, and five feelings of increased/decreased alertness and concentration.

## 7. Discussion

This is the first systematic review to assess the effects of non-invasive neuromodulation treatments in HD. Three types of interventions (ECT, TMS, and tDCS) were investigated in HD, and most studies were case reports or series, with only four RCT. Eighteen out of nineteen studies showed improvement in HD symptoms, suggesting that neuromodulation might play a role in the clinical management of the disease [17,18,19,20,21,22,23,24,25,26,27,28,29,30,31,33,34,35]. However, studies were very heterogeneous in terms of neuromodulation parameters and clinical assessments. Regarding the latter, for example, only a handful of studies reported side effects and tolerability [18,22,23,27,30,33,34]. In this context, it is difficult to draw definite conclusions about the effectiveness or superiority of any particular neuromodulation modality for HD-related symptoms due to the limited quality of the available studies.

Electroconvulsive therapy (ECT) is one of the oldest and long-standing treatment modalities for the management of severe psychiatric disorders [36]. The mechanisms of action of ECT are thought to involve short-term reorganization of brain connectivity and network functioning, as well as long-term effects involving neuroplasticity [13]. Although the specific mechanisms by which ECT can alleviate behavioral symptoms are not completely elucidated, current literature suggests effects on a range of neurobiological systems (e.g., glucose metabolism, blood flow, oxygen consumption) and modulation of brain networks by influencing their connectivity [37]. Other important mechanisms of action of ECT involve the modulation of hypothalamic and pituitary activity, long-term neurotrophic effects, and reduction in inflammation [37,38]. The major indication for ECT is treatment-resistant major depressive disorder, while other clinical conditions include catatonia, and severe manic and psychotic episodes [39,40]. Of the 27 patients with HD included in the 13 ECT studies, 26 had improved behavioral symptoms, mainly symptoms of depression and psychosis. Given that the ECT parameters used in the different studies were highly heterogeneous, it was not possible to identify specific protocols responsible for enhanced improvement of determined behavioral domains in HD. Of note, depressive [17,19,23,25,27,28] and psychotic symptoms [17,18,22,23,24,27,29] were the ones most frequently reported as ameliorated, which is in line with the general psychiatry literature on ECT [36,39,40].

The current understanding of the pathophysiology of motor symptoms in HD points towards a disorganized activity of the sensory-motor network and altered neurotransmission between the motor cortex and basal ganglia [41,42,43]. ECT can influence the connectivity between cortical areas and deep brain structures, such as the thalamus and basal ganglia, thereby influencing multiple brain networks [13]. Seven ECT studies described improvement in motor symptoms such as chorea and ambulation, while three studies reported worsening of motor symptoms (e.g., increase in involuntary movements) and four studies did not report any changes. Therefore, the effect of ECT on chorea and other motor symptoms is controversial.

Cognitive side effects are commonly reported after ECT, but they are usually transitory and sustained cognitive impairment is rare [44]. Interestingly, increased cognitive performance after ECT has been reported in some studies with other populations (e.g., patients with major depression) [45,46,47]. Four out of the thirteen ECT studies described an improvement in cognition, as assessed by cognitive screening tools (MMSE and MoCA) [18,19,23,28], and one study reported improvement through clinical judgement [27]. Only Ranen et al. [18] reported a case of non-transitory cognitive deterioration. It is worth noting that ECT, besides acutely influencing neural networks, has long-term effects on neuroplasticity and neurogenesis, which can explain the positive effects on cognition in a few studies [13]. Given the potential of cognitive side effects with ECT, and the progressive nature of cognitive decline in HD, these latter findings must be seen with caution, and the cognitive safety of ECT in this population remains to be determined.

In summary, despite the shortcomings of the available studies, ECT can be seen as a potential tool in the management of behavioral symptoms of HD, especially concerning treatment resistant depression and psychosis. The high heterogeneity and low number of studies make it impossible to compare efficacy and safety between ECT protocols. It is important to emphasize that the safety of ECT, especially for motor and cognitive symptoms, is not clearly established in HD. This urges new studies to provide complete descriptions of ECT parameters, as well as the use of validated instruments to precisely quantify improvements and potential side effects. New ECT studies should also incorporate structural, network, and brain connectivity markers. This would contribute to a better understanding of the role of ECT in the treatment of HD, also providing invaluable insights on the pathophysiology of the related symptoms.

TMS is a non-invasive procedure that works on the principle of Faraday’s law of electromagnetic induction that an electric current can generate a magnetic field. This magnetic field can induce neuronal depolarization, excitation, or inhibition by generating an action potential in certain parts of the brain [48]. TMS is currently FDA approved for treatment resistant depression [49], and has been investigated in different clinical contexts, such as migraine with aura [50], tinnitus [51], obsessive-compulsive disorder [52], attention deficit hyperactivity disorder (ADHD) [53], and smoking cessation [54].

Three out of four studies with TMS reported an improvement in symptoms in patients with HD. Brusa et al. and Shukla et al. [30,32] used a figure-of-eight coil positioned at the supplementary motor area and applied 900 pulses at 1 Hz frequency, and intensity was set at 90% of the motor threshold. While Brusa et al. reported motor improvement with just one session, Shukla et al. interrupted the intervention after seven sessions due to the lack of improvement. One of the issues of TMS is the heterogeneity in target accuracy, which is associated with a range of factors, including anatomical and functional variations [55,56,57,58], that might have contributed to the conflicting results. Other contributing factors might involve coil size (the two studies differed by 10 mm), and pulse shape and current direction that were not reported in these studies. It is also worth noticing that the improvement reported by Brusa et al. was transient. The study of Groiss et al. [31] reported a mixed effect of high frequency rTMS on motor performance in the ipsilateral and contralateral hands of patients with HD. This effect was not observed in healthy individuals, and may actually reflect altered cortical neuroplasticity and altered neural networks in HD. Groiss et al. also reported an antidepressant effect of low frequency rTMS that lasted for at least two weeks. Davis et al. [59] reported an antidepressant effect of dTMS, a different modality of rTMS that can affect deeper brain structures (i.e., up to 5.5 cm from the coil). The improvements were sustained for at least eight months after a course of 49 daily sessions. Only two studies reported mild side effects from TMS [30,33], which are in line with the literature [60,61].

The application of TMS for neuropsychiatric disorders is a growing field. The results from the studies reviewed here, in addition to the low incidence of side effects of TMS, point to a possible role for this intervention in the treatment of HD and investigation of underlying mechanisms. Traditional rTMS protocols, such as the ones in the selected studies, are currently used in clinical settings for major depression, but newer TMS protocols, such as theta burst TMS, are gaining momentum and may be interesting alternatives for future studies in HD [62]. Future studies are also warranted to better understand optimized protocols that can impact specific domains of HD symptoms. A major drawback of TMS is the need of multiple and/or frequent visits to clinics for its application [63,64]. In this sense, new studies should also consider strategies with shorter time frames to alleviate patient burden and treatment costs.

tDCS is a modality of tES that involves application of a low voltage, direct electric current to the scalp through two or more battery powered electrodes embedded in a sponge soaked with saline [65]. Frequently induced stimulation criterion varies from 1 to 2 mA in current intensity, from 3.5 to 100 cm^2^ in electrode size, and from 5 to 20 min stimulation time in most of the studies [66]. The placement of the electrodes depends on the objective of the study or treatment. In conventional tDCS, the anode is placed at the left dorsolateral prefrontal cortex, and the cathode at the right supraorbital area [67]. When compared to ECT and TMS, tDCS has relatively fewer adverse effects, is cheaper and very portable, making it feasible to be used at home with remote supervision [64,68,69]. tDCS has been studied for different clinical applications such as major depression [68], chronic pain [70], cognitive augmentation [71], and neurorehabilitation [72]. The studies involving tDCS in HD have not assessed any behavioral symptoms, but reported improvement in cognition (i.e., working memory) [34] and motor symptoms [35]. Bocci et al. [35] showed an improvement in motor symptoms of HD with a course of anodal cerebellar tDCS. The role of the cerebellum in the mechanisms of HD pathophysiology and symptomatology is still not fully understood. However, a growing body of evidence supports a role for the cerebellum in motor, cognitive, and behavioral symptoms of HD [73], specifically the disrupted connectivity between the cerebellum and the basal ganglia. The potential of tDCS to positively influence brain connectivity has been demonstrated in other clinical settings [74,75]. It is possible that the results from Bocci et al. may reflect beneficial effects of cerebellar tDCS on the connectivity between the cerebellum and the basal ganglia. Eddy et al. used anodal tDCS over the DLPFC of HD patients and reported cognitive improvement in measures of working memory [34]. These findings are in line with the literature on tDCS and cognition showing positive influence of tDCS on executive functions, mainly working memory and attention [76].

The current review has limitations, mostly reflecting the relatively small number and quality of the available studies. It was not possible to perform a quantitative synthesis of the data due to significant differences in intervention type, stimulation parameters, outcome measures, and most studies encompassed case reports or case series. In addition, there was poor evidence of blinding of staff delivering treatment in all RCT studies [30,31,34,35] and of participants in one study [30], with most studies having short follow-up period (2–4 weeks). Therefore, more rigorous and robust studies of noninvasive neuromodulation in HD with representative sample, proper randomization, blinding, and assessment alongside adequate follow-up are needed to confirm (or refute) these preliminary findings.

To summarize, due to the limited methodological quality of most studies, it is not possible to draw definite conclusions about the effects of any particular neuromodulation treatment on HD-related symptoms. The most studied non-invasive neuromodulation modality in the context of HD was ECT, and the reports suggest that this intervention might play a role in the management of HD-related depression and psychosis. Safety issues related to ECT for this population remain to be defined. There are promising preliminary reports of efficacy without major adverse effects of TMS and tDCS for HD-related symptoms, and further investigations are definitely warranted.

## Figures and Tables

**Figure 1 jcm-12-02002-f001:**
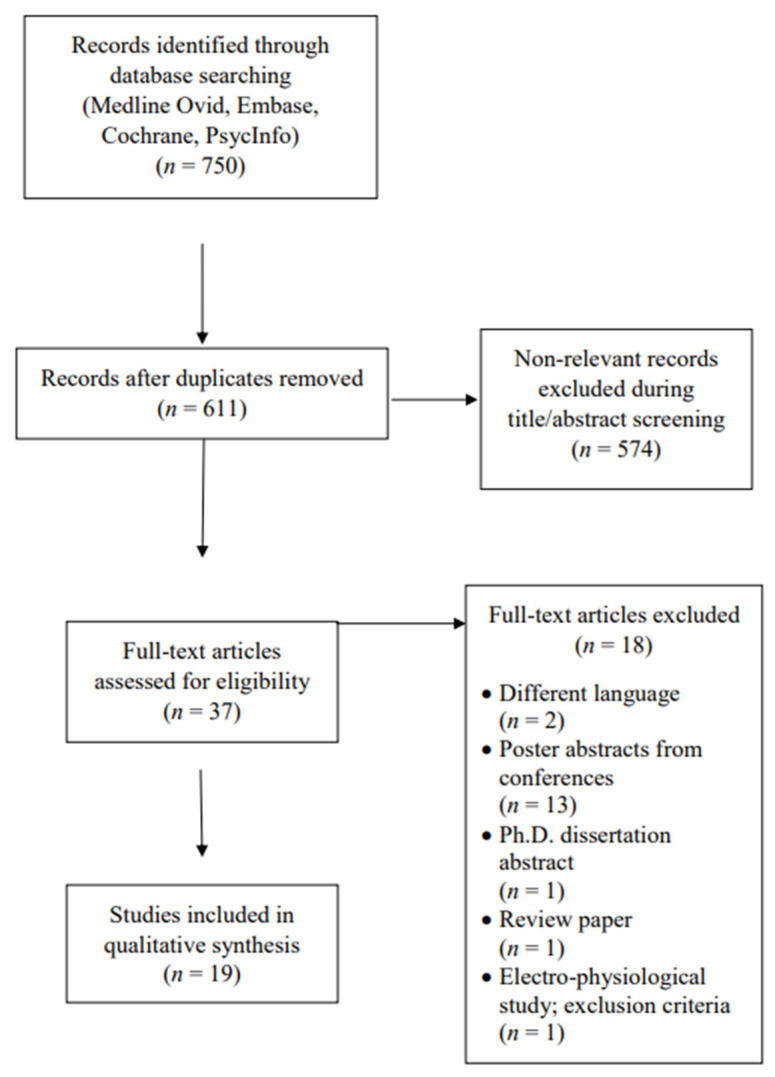
PRISMA Flow Diagram.

**Table 1 jcm-12-02002-t001:** Studies investigating electroconvulsive therapy (ECT) in patients with Huntington’s disease. RUL = right unilateral; BL = bilateral; MMSE = Mini-Mental State Exam; BFCRS = Bush-Francis Catatonia Rating Scale; Ham-D = Hamilton Depression Rating Scale; PANSS = Positive and negative symptoms scale; BPRS = Brief psychiatric rating scale; MADRS = Montgomery-Åsberg Depression Rating Scale; UHDRS = Unified Huntington’s Disease Rating Scale; MoCA = Montreal Cognitive Assessment; Q-LES Q-SF = Quality of Life Enjoyment and Satisfaction Questionnaire-Short Form; CGI = Clinical Global Impression; NA = Information was not available in the article; ↓: Symbol indicating reduction; ↑: Symbol indicating increase.

Author, Date	Study Type	Sample	Intervention	Tools	Outcome	Side Effects
Evans et al. (1987) [17]	Case—Report	1 (F, 49 years old) CAG repeat: NA TFC: NA	**Intervention:** ECT, parameters not specified. **Electrode position:** RUL. **Number of sessions:** 6 sessions. **Seizure length:** 239 s by EEG recordings.	Physician examination and observations (symptoms: cognitive functional, and behavioral)	- There was improvement in psychotic symptoms. - By the end of the sixth ECT session, the patient’s auditory hallucinations had ceased, and her depressive symptoms had subsided. - Suicidal and homicidal tendencies lessened after the second session.	Mild increase in choreiform movements during the course of treatment.
Ranen et al. (1994) [18]	Case—Series	6 (2F, 4M, aged between 41 and 62 years) CAG Repeat: NA TFC: NA	**Intervention:** ECT, initial parameters with 70 Hz, pulse width of 1.0 ms, and duration of 2.0 ms. Parameters were increased as necessary to achieve adequate seizure duration. **Electrode position:** Initial treatment with RUL for all patients. BL placement was used when necessary to achieve adequate seizure duration. Only RUL was used in Cases 1 and 2; Case 4 used RUL and BL in different sessions; Cases 3, 5 and 6 were not specified. **Number of sessions:** 7 (Case 1), 7 (Case 2), 8 (Case 3 and Case 6), 5 (Case 4), and 9 (Case 5). **Seizure length:** Raged from 5 to 120 s.	Physician observations (symptoms: alteration in movements, behavior, psychiatric, and cognitive functions) Mini-Mental State Exam (MMSE)	- Complete resolution of delusions (Cases 2 and 6). - All Cases, except Case 3, experimented some improvement. - Apathy fared worse than other symptoms. - MMSE improved in Case 2 and 4 while it worsened in Case 3.	One patient developed psychosis and cognitive impairment (Case 3), and one patient had worsened catatonia with agitation (Case 4).
Lewis et al. (1996) [19]	Case—Report	1 (M, 65 years old) CAG Repeat: 44 TFC stage: Il/V	**Intervention:** ECT, 90 Hz, brief pulse 1.0–1.6 ms, dynamic energy 33.3–55.7 joules. **Electrode position:** Frontotemporal (BL). **Number of sessions:** 8 sessions. **Electrode position:** frontotemporal (bilateral) **Seizure length:** Range between 24 and 140 s.	Hamilton Depression Rating Scale (Ham-D) and MMSE	- ↓ Ham-D score from 36 to 10 after the eighth sessions. - ↑ MMSE score from 23/30 to 24/30 after the fifth sessions—posttreatment MMSE score was not available.	NA
Beale et al. (1997) [20]	Case—Report	1 (M, 56 years old) CAG Repeat: 46 TFC: NA	**Intervention:** First ECT session: stimulation dose started at 12.7 joules (72 mC), which did not cause seizure, and the dose was increased to 40.1 joules (229 mC). Second ECT session: stimulation dose was at 40.8 joules (233 mC). Stimulus dose was increased by 10–20 joules for each subsequent session. **Electrode position:** BL **Number of sessions:** 11 ECT, given 3 times per week. Maintenance ECT given once every 3 months. **Seizure length (average):** 34.3 s (motor) and 39.1 s (EEG). Range: 19–57 s.	Self-reported and clinical examination	- Mild improvement was noted immediately after the last session. - At 6 weeks after treatment, the patient had significant improvement in his movement, with a reduction in nocturnal chorea. - ↑ Weight (10 lbs) was noted. - The improvement in motor symptoms was sustained for the next three years after the treatment.	NA
Merida-Puga et al. (2011) [21]	Case—Report	1 (F, 26 years old) CAG Repeat: 45 TFC: NA	**Intervention:** ECT, stimulus was calculated using the half-life method and increased by up to 25%. **Electrode position:** NA **Number of sessions:** 42 sessions of ECT, (13 in the first, 15 in the second, and 14 in the third cycle of intervention). **Seizure length:** 42 to 80 s.	Self-reported Bush-Francis Catatonia Rating Scale (BFCRS)	- ↓ BFCRS from 26 (1st admission) to 4 after 124 days of hospitalization. - The patient could partially take care of herself to a certain extent but required assistance with bathing and eating at times. - ↓ verbal output and partial withdrawal.	NA
Nakano et al. (2013) [22]	Case—Report	1 (M, 59 years old) CAG Repeat: 44 TFC: NA	**Intervention:** modified ECT, stimulation parameters not specified. **Electrode position:** NA **Number of sessions:** 21 sessions of modified ECT over a period of 6 months’ time. **Seizure length:** NA	MMSE scores, positive and negative symptom scale (PANSS), and brief psychiatric rating scale (BPRS)	- ↓ PANSS from 139 to 68, and BPRS from 84 to 36. - Improvement of delusions and hallucinations after the fourth treatment with mECT without worsening involuntary movements. - When compared to the analysis before mECT, the amount of 99mTc uptake in the basal ganglia, cingulate gyrus, and thalamus was considerably reduced in the single-photon emission computed tomography (SPECT) scan following mECT therapy.	Transient cardiovascular problem and anterograde/retrograde amnesia
Cusin et al. (2013) [23]	Case—Series	7 (4F, 3M, age range from 20 to 56) CAG Repeat: NA TFC: NA	**Intervention:** ECT, pulse width of 1 msec, frequency of 90 Hz, and duration of 2–4 s. **Electrode position:** RUL. **Number of sessions:** 4 to 13 sessions to treat the acute series. One patient had further ECT to treat depressive recurrence several months later, and two patients required maintenance ECT at regular intervals to maintain the recovery. **Seizure length:** NA	Self-reported and clinical examination (symptoms: motor and behavioral symptoms)	- Remission of suicidal thoughts in four patients. - Mood was improved in six patients. - Psychosis had improved in three patients. - ↑ cognitive abilities in one patient. - Improved cooperation with treatment was noted in three patients. - Ambulation had improved in five patients.	Two patients developed short-term agitation upon recovery from anesthesia.
Magid et al. (2014) [24]	Case—Report	1 (F, 57 years old) CAG Repeat: 43 TFC: NA	**Intervention:** ECT, brief-pulse stimulus with age-based stimulus dosing. **Electrode position:** BL (bi-temporal). **Number of sessions:** 4 sessions before hospital discharge. Maintenance ECT once every 1 to 4 weeks for 6 months. **Seizure length:** NA	Self-reported and clinician observation (symptoms: visual hallucinations, agitation, frequent verbal outbursts, weight loss, impaired judgement, disorientation, and the patient had become non-communicative).	- Patient’s psychosis resolved. - Improvement in appetite, weight, and physical heath after the fourth ECT treatment, which allowed the patient to be discharged home to her family’s care.	NA
Petit et al. (2016) [25]	Case—Report	1 (M, 60 years old) CAG Repeat: 41 TFC score: 2	**Intervention:** ECT, stimulation parameters not specified. **Electrode position:** NA **Number of sessions:** 18 sessions. **Seizure length:** NA	Expanded version of the Brief Psychiatric Rating Scale (BPRS-E); Clinical Global Impression (CGI); Montgomery-Åsberg Depression Rating Scale (MADRS); Unified Huntington’s Disease Rating Scale (UHDRS); MMSE	After 12 ECT sessions: - UHDRS scores: ↓ motor symptoms (from 47/124 to 37/124), behavioral (from 54/88 to 26/88), and functional assessment (from 41/50 to 36/50), and ↑ independence (from 45/100 to 60/100) and total functional capacity (from 2/5 to 3/5). - ↓ MADRS (from 47/60 to 7/60), BPRS-E (from 88/168 to 38/168) and CGI (from 6/7 to 5/7). After 1 year: - UHDRS scores: Moderate ↑ motor symptoms (from 47/124 to 57/124), functional assessment (from 41/50 to 42/50), and independence (from 45/100 to 55/100) and total functional capacity (from 2/5 to 4/5).	NA
Shah et al. (2017) [26]	Case—Report	1 (F, 51 years old) CAG Repeat: NA TFC: NA	**Intervention:** ECT, stimulation parameters not specified. **Electrode position:** BL (bitemporal). **Number of sessions:** 5 sessions. **Seizure length:** NA	Self-reported and clinical examination	- Improvement in patients’ behaviors and interaction (↓ irritability, and verbal agitation, and no episodes of physical agitation) after the third ECT treatment. - The clinical improvements remain for two months, but patients’ agitation persisted afterward.	NA
Adrissi et al. (2019) [27]	Case—Series	4 (1F, 3M age range from 38 to 52) CAG Repeat: 44 (case 1), 42 (case 2), 46 (case 3) and 39 (case 4) TFC: NA	**Intervention:** ECT, Case 1: 0.50 ms pulse width, 40% charge dose. Case 2: 0.25 ms pulse width, 10–70 Hz frequency, 5–25% charge. Case 3: 0.25–1.0 ms pulse width, 40–140 Hz frequency, 100% charge dose (except for two sessions with 50%). Case 4: 0.25 ms pulse width, 40 Hz frequency (except initial session 10 Hz), 25–50% charge dose (except for first session with 5%). **Electrode position**—BL (bitemporal) (Cases 1 and 3) and RUL (Cases 2 and 4). **Number of sessions:** 29 (Case 1), 27 (Case 2), 41 (Case 3), 7 (Case 4). **Seizure length**—ranged from 21 to 84 s.	Symptoms: Psychiatric—suicidal ideation, anxiety, depression, agitation evaluated using clinical examination, motor—evaluated using UHDRS—case 1 and 2 and cognition evaluated using Montreal Cognitive Assessment (MoCA)—case 2	- All patients improved in mood and suicidal thoughts after the ECT course. - Both individuals with co-existing psychosis improved in terms of psychotic symptoms. - One patient experienced additional subjective improvement in motor function.	One patient developed irritability and delirium after ECT
Abeysundera et al. (2019) [28]	Case—Report	1 (F, 56 years old) CAG Repeat: NA TFC: NA	**Intervention:** ECT, stimulation parameter not specified. **Electrode position:** RUL. **Number of sessions:** 10 sessions. **Seizure length:** NA	MADRS; MoCA; Quality of Life Enjoyment and Satisfaction Questionnaire-Short Form (Q-LES Q-SF); UHDRS.	- Subjective improvement in mood from a pre-treatment score of 2/10 (1 = worst, 10 = best) to 6/10 after ECT sessions number 4. - In the follow-up evaluation, the patient mood deteriorated again after one month of the end of the treatment. - ↓ MADRS from 45/60 to 23/60; - ↑ MOCA from 21/30 to 25/30, Q-LES Q-SF 38/100% to 59/100%, and UHDRS from 26/124 to 46/124.	After fifth session of ECT, there was worsening of her involuntary movements along with increasing difficulty with walking. After 10 ECT sessions, the patient experienced worsening balance and involuntary movements.
Mowafi et al. (2021) [29]	Case—Report	1 (F, 57 years old) CAG Repeat: 46 TFC: NA	**Intervention:** ECT, current intensity: First 12 sessions used between 75 and 150 mC; last 12 sessions used between 150 and 225 mC. Maintenance ECT used between 150 and 300 mC. **Electrode position:** BL. **Number of sessions:** 24 ECT sessions and later maintenance ECT once every week and later fortnightly (total number of sessions is not described). **Seizure length:** from 30 to 60 s.	Self-reported and clinical examination	- Remission of patient’s psychomotor retardation, Psychotic symptoms persisted in the form of command hallucinations, after the first set of 12 ECT sessions. - After a second set of 12 ECT sessions, there was a partial remission of the auditory hallucinations, which had reduced in frequency and no longer distressed the patient. - Delusion of guilt resolved. - Appetite improvement after 10th ECT session.	NA

**Table 2 jcm-12-02002-t002:** Studies investigating transcranial magnetic stimulation (TMS) in patients with HD. AIMs = Abnormal Involuntary movement scale; RT = Reaction times; DST = Digit span test; BDI = Beck Depression Inventory; HD-ADL = HD-activity of daily living; TFC = Total Functional Capacity; GDS = Geriatric Depression Scale; ↓: Symbol indicating reduction; ↑: Symbol indicating increase.

Author, Date	Study Type	Sample	Intervention	Tools	Outcome	Side Effects
Brusa et al. (2005) [30]	Clinical trial	4 (sex and age NA) CAG Repeat: NA TFC stage: II to IV	**Intervention (crossover design):** **Sham rTMS**—consisted of 900 pulses delivered at 1 Hz, intensity was set at 90% of the resting motor threshold (RMT), and the coil was angled away so that no current was induced in the brain. **1 Hz rTMS**—consisted of 900 pulses delivered at 1 Hz, intensity was set at 90% of the resting motor threshold. **5 Hz rTMS**—consisted of 18 trains of 50 stimuli at 5 Hz frequency separated by 40 s of pause, delivered at 110% resting motor threshold for a total of 900 pulses. **Frequency**—one-time stimulation. **Coil type/position**—Figure-of-eight coil/supplementary motor area (SMA) of both hemispheres (3 cm anterior to Cz in the sagittal midline). **Interval period**—The patients received each of the stimulation in three consecutive days (one day for each stimulation i.e., sham, 1 Hz and 5 Hz)	Abnormal Involuntary movement scale (AIMs), UHDRS—chorea and bradykinesia items of the motor section.	**1 Hz rTMS** - ↓ AIMs in all patients. - The chorea score on UHDRS decreased only 15–30 min post stimulation. **5 Hz rTMS** - No beneficial effect was induced.	One patient’s bradykinesia transiently worsened immediately after receiving 1 Hz rTMS stimulation.
Groiss et al. (2012) [31]	Clinical trial	8 (4F and 4M, age range from 32 to 63) CAG Repeat: ranges from 39 to 51. TFC: NA	**Intervention (crossover design):** **Sham rTMS**—consisted of 10 trains of 5 Hz rTMS with duration of 4 s and intertrain interval of 60 s and a total number of 200 stimuli were applied during one session. **1 Hz rTMS**—consisted of a train of 200 stimuli applied at 1 Hz, total number of 200 stimuli were applied during each session. **10 Hz rTMS**—consisted of 10 trains of 10 Hz rTMS with duration of 2 s and intertrain interval of 60 s and 200 stimuli were applied during each session. **Coil type/position:** Figure-of-eight coil/The coil was positioned on the scalp over the left primary motor cortex (M1). **Frequency:** one-time stimulation **Interval period**—three sessions separated by at least two weeks.	UHDRS; Nine Hole Peg Test (NHPT); Reaction times (RT), including simple (sRT) and choice (cRT)); Digit span test (DST); Beck Depression Inventory (BDI); HD-activity of daily living (HD-ADL).	**1 Hz rTMS** - Improvement on mood for at least one week after intervention. **10 Hz rTMS** - sRT of the contralateral hand was prolonged immediately and one hour after stimulation. - In the cRT task, there was a shortened reaction time for the ipsilateral hand immediately, one day and two weeks after the intervention.	NA
Shukla et al. (2013) [32]	Case-Series	2 (sex and age NA) CAG Repeat: NA TFC score: 1 (case 1), 2 (case 2).	**Intervention:** rTMS, 900 pulses at 1 Hz frequency and intensity was set at 90% of the motor threshold **Coil type/position:** Figure-of-eight coil/supplementary motor area of both cerebral hemispheres, which is approximately three cm anterior to the Cz in the sagittal midline. **Frequency:** Seven sessions on a once-daily basis **Interval period:** 1 day apart (7 consecutive sessions in total).	AIMs Total Functional Capacity (TFC)	- No improvement was observed through the course or after the treatment.	NA
Davis et al. (2016) [33]	Case-Report	1 (M, 77 years old with late onset HD, TRD and GAD) CAG Repeat: NA TFC: NA	**Intervention:** dTMS, 1600 pulses at 1 Hz frequency at 120% of the motor threshold. **Coil type/position:** H coil/right dorsolateral prefrontal cortex. **Frequency:** 49 days (once per day).	The Geriatric Depression Scale (GDS)	- ↓ GDS score from 14/15 to 2/15. - The remission remains after eight months without maintenance treatments. - Self-reported improvement of cognitive impairments, anxiety, and physical pain.	Lacrimation in the right eye, as well as scalp discomfort at the treatment site.

**Table 3 jcm-12-02002-t003:** Studies investigating transcranial direct current stimulation (tDCS) in patients with HD. DOT-A = Digit ordering test-adapted; ↓: Symbol indicating reduction; ↑: Symbol indicating increase.

Author, Date	Study Design	Sample	Intervention	Outcome Measures	Outcome	Side Effects
Eddy et al. (2017) [34]	Clinical Trial	20 (Sex: NA; age range between 50 and 72) CAG Repeat: NA TFC: NA	**Intervention: Anodal tDCS**—1.5 mA tDCS was sustained for 15 min, followed by a 60- second ramp down. **Sham**—after the 60 s ramp up, stimulation was programmed to ramp down again. **Electrode position:** The anode was placed over F3 to stimulate left DLPFC and the cathode was placed over the contralateral orbital area (FP2). **Duration:** 17 min/session. **Frequency:** 1 day for anodal tDCS and 1 day for sham. **Washout period:** One week.	Outcome measures (pre and post-tests): Digit ordering test-adapted (DOT-A), Stroop test, 1-back and 2-back tests (N-back tasks);	Primary: Working Memory (WM). **DOT-A** - On average, WM span as measured by the DOT-A task increased by half a point from pre- to post-intervention for the tDCS condition. N-back tasks: **1-back test** - Not a significant change because baseline scores were already high. **2-back test** - Significant improvement from pre- to post-intervention for tDCS, but not for sham. **Stroop task** - Both tDCS and sham conditions were associated with faster post intervention performance compared to pre-intervention test. Additional outcome: **Tolerability** - There were no dropouts and no evidence that side-effects were more common with tDCS than sham (no reported effects: anodal tDCS *n* = 7; sham *n* = 7).	Tingling (anodal tDCS *n* = 6; sham *n* = 6), itching (anodal tDCS *n* = 5; sham *n* = 3), feelings of increased/decreased alertness and concentration (anodal tDCS *n* = 5; sham *n* = 5).
Bocci et al. (2020) [35]	Clinical Trial	4 (2M, 2F, aged between 43 and 50 year) CAG Repeat: ≥40 UHDRS motor score: >5 TFC score: >7	**Interventions:****Anodal tDCS**—2.0 mA tDCS was sustained for 20 min, followed by a decrease in current in a ramp-like manner; current intensity: ∼0.08 mA/cm2. Sham—the current was turned on for 5 s and then turned off in a ramp-shaped fashion, thus inducing skin sensations similar to those produced by real tDCS. **Electrode position**—The anode was applied on the median line, 2 cm below the inion, with lateral borders about 1 cm medially to the mastoid apophysis, and the cathode over the right shoulder. **Duration** 20 min per session for 5 consecutive days (3 months interval between interventions).	Tools: UHDRS-part I. Time points: Baseline (T0), end of the stimulation week (T1), and 4 weeks later (T2).	- Anodal tDCS effect: - ↓ UHDRS-I over time (T1 and T2 vs. T0) (*p* <0.01) and when compared to sham at T1 (*p* = 0.46) and T2 (*p* = 0.48). - ↓ UHDRS-1 dystonia sub scores over time (*p* = 0.04) and when compared to sham at T1 (*p* = 0.46) and T2 (*p* = 0.48). - Trend to ↓ UHDRS-1 chorea sub scores over time (*p* = 0.07). - ↓ UHDRS-I score at T1 and T2 compared to Sham.	NA

## Data Availability

Not applicable.

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
