# Peer review of "Non-Invasive Neuromodulation Methods to Alleviate Symptoms of Huntington’s Disease: A Systematic Review of the Literature"

_jcm, 2023, doi:10.3390/jcm12052002_

Round 1

Reviewer 1 Report (Previous Reviewer 2)

Thank you for giving me the opportunity to review this manuscript.

I think it is necessary to revise the manuscript.

It is very important to say that neuromodulation cannot improve the etiology of Huntington's Disease. Actually, this disease is caused by a mutation in the gene for a protein called huntingtin. This causes the building blocks of DNA called cytosine, adenine, and guanine to repeat many more times. This causes gradual degeneration of parts of the basal ganglia called the caudate nucleus and putamen. However, neuromodulation has never been shown to improve this genetic mutation nor the degeneration.

1) Please change the title as "Non-invasive neuromodulation methods to alleviate symptoms in patients with Huntington's disease: a systematic review of the literature"

2) Please change the sentences of " From the 27 patients included in the 13 ECT studies, 26 had improvement of psychiatric symptoms, highlighting the potential of ECT in the management of HD patients." as "From the 27 patients included in the 13 ECT studies, 26 had improvement of psychiatric symptoms, highlighting the potential of ECT to alleviate depression and psychosis HD patients." Furthermore, please add that the safety of ECT in motor and cognitive symptoms has never been established.

3) Please delete the sentene of "The ECT potential to positively impact sensorimotor networks makes it a putative method in the treatment of HD." That is because the safety of ECT on motor symptoms in patients with HD has never been established. Furthermore, ECT do not always change the degeneration of parts of the basal ganglia called the caudate nucleus and putamen. Please add this point in the discussion and limitations.

4) Please delete the sentences of "Although seemingly counterintuitive, increased cognitive performance after a course of ECT has been reported in studies with other populations with cognitive impairment [46-48]. It is worth emphasizing that ECT, besides acutely influencing neural networks, has long term effects on neuroplasticity and neurogenesis, what can explain the observed effects on cognition [13]." The effects of ECT on cognition are too overestimated. Please avoid using those sentences. Instead, please add the sentences to emphasize that the effects on cognition have been controversial.

5) Please change the sentences of "Given the clinically meaningful results, despite the shortcomings of the available studies, ECT can be seen as an important tool in the treatment of HD, especially concerning treatment resistant depression and psychosis." as "ECT can be seen as a potential tool to improve depression and psychosis in patients with HD".

It is necessary to revise the manuscript to avoid overestimation of the effects of ECT on some symptoms in patients with HD.

Author Response

Reviewer: 1

It is very important to say that neuromodulation cannot improve the etiology of Huntington's Disease. Actually, this disease is caused by a mutation in the gene for a protein called huntingtin. This causes the building blocks of DNA called cytosine, adenine, and guanine to repeat many more times. This causes gradual degeneration of parts of the basal ganglia called the caudate nucleus and putamen. However, neuromodulation has never been shown to improve this genetic mutation nor the degeneration.

1) Please change the title as "Non-invasive neuromodulation methods to alleviate symptoms in patients with Huntington's disease: a systematic review of the literature"

Response: The reviewer brings up an important view. We agree that neuromodulation cannot improve the causal mechanisms of Huntington’s disease (HD). This is inherent to the mechanisms of neuromodulation interventions as well as the disease. We used the word “treatment” in reference to the set of measures aiming to ameliorate the clinical manifestations of HD. We understand the reviewer’s concerns and changed the title accordingly. Please see the title of the new version of the manuscript.

2) Please change the sentences of " From the 27 patients included in the 13 ECT studies, 26 had improvement of psychiatric symptoms, highlighting the potential of ECT in the management of HD patients." as "From the 27 patients included in the 13 ECT studies, 26 had improvement of psychiatric symptoms, highlighting the potential of ECT to alleviate depression and psychosis HD patients " Furthermore, please add that the safety of ECT in motor and cognitive symptoms has never been established.

Response: We modified the sentence. Of note, depression and psychosis were the most frequently improved psychiatric symptoms, but improvement of other related manifestations such as suicidality, catatonia and agitation were also reported. We highlighted the evidence of ECT effects on depression and psychosis at the end of the paragraph (see lines 265-272). Regarding the safety of ECT in relation to cognitive and motor symptoms, we tried to better discuss these points in the new version of the manuscript (see lines 298-300).

3) Please delete the sentence of "The ECT potential to positively impact sensorimotor networks makes it a putative method in the treatment of HD." That is because the safety of ECT on motor symptoms in patients with HD has never been established. Furthermore, ECT do not always change the degeneration of parts of the basal ganglia called the caudate nucleus and putamen. Please add this point in the discussion and limitations.

Response: Although we did not claim that ECT was capable of changing the degeneration of specific subcortical structure (i.e. basal ganglia), we agree that this sentence should be reformulated. Please see the changes made on the new version of the manuscript (see lines 275-277).

4) Please delete the sentences of "Although seemingly counterintuitive, increased cognitive performance after a course of ECT has been reported in studies with other populations with cognitive impairment [46-48]. It is worth emphasizing that ECT, besides acutely influencing neural networks, has long term effects on neuroplasticity and neurogenesis, what can explain the observed effects on cognition [13]." The effects of ECT on cognition are too overestimated. Please avoid using those sentences. Instead, please add the sentences to emphasize that the effects on cognition have been controversial.

Response: We agree that these sentences might mislead to an overestimate of the effect of ECT in cognition. Actually, the evidence of neuroplastic and neurotrophic effects of ECT, as well as the cognitive improvement (not acutely but in the long term) observed in different populations exposed to the same intervention, are of relevance to the discussion of the findings from the selected studies. We rephrased these sentences in the new version of the manuscript also discussing the controversial results of ECT effects on cognition (see lines 282-285 and 291-294).

5) Please change the sentences of "Given the clinically meaningful results, despite the shortcomings of the available studies, ECT can be seen as an important tool in the treatment of HD, especially concerning treatment resistant depression and psychosis." as "ECT can be seen as a potential tool to improve depression and psychosis in patients with HD".

Response: We agree with the reviewer that these sentences should be reformulated. See the proposed changes in the lines 295-297 of the new version of the manuscript.

It is necessary to revise the manuscript to avoid overestimation of the effects of ECT on some symptoms in patients with HD.

Response: As requested, we toned down the statements on the potential benefits of ECT for HD (see new Abstract and Discussion).

Reviewer 2 Report (Previous Reviewer 1)

I think that the authors took my comments earnestly and revised the manuscript. The problems I pointed out have been sufficiently improved.

Author Response

Reviewer 2: I think that the authors took my comments earnestly and revised the manuscript. The problems I pointed out have been sufficiently improved.

Response: Thank you for the positive comment on our revised version.

Reviewer 3 Report (New Reviewer)

-Abstract – it is really too stressful sentences such as “Despite eighteen out of nineteen studies showed improvement in HD symptoms, further investigations are required due to the quality of the available evidence and related methodological shortcomings. It is likely that HD-related symptoms can be treated with neuromodulation methods with little or controllable side effects.” Since authors are covering different psychological, cognitive and motor symptoms, it is difficult to determine what conclusions are related to what. The results can be presented as the efficacy of different studies in treating different symptoms. Which symptoms are more likely to be treated efficiently with neuromodulation techniques? What is the duration of the therapeutic effects are there any information on this? The reader really does not have a clear take-off message. Which symptoms are mostly studied and can be investigated therapeutically with what?

-In the introductory under the aims, the terms motor, cognition, and behavior-related symptoms are used. Later in the manuscript, the authors use “psychiatric symptoms”. It is suggested to introduce the proper terms and to be consistent with the presentation of the symptomatology versus testing outcomes (later presented in the Results section. In the introduction, the authors can introduce examples of symptomatology under superordinate terms such as “psychiatric symptoms,”  or psychological symptoms, or behavioral symptoms, or motor symptoms.

-Under the searching strategy, the authors included TMS studies, but later on when presenting the authors can specify which TMS device was used: TMS with EMG without navigation, line navigate TMS or e-field navigated TMS?

- What is the difference between rTMS and deep TMS? RTMS refers to stimulation protocol but deep refers to coil usage right? It is strange to use rTMS vs dTMS please check this. I believe H coil was related to author's explanation of dTMS, if yes this need to be corrected.

-Please state in the introductory part that non of the neuromodulation technique does not have FDA clearance used for treatment, the use of neuromodulating techniques in HD are still purely research-based.

-Raw 105 “The most common reason for exclusion was not using neuromodulation as a treatment modality for HD”. Can authors explain which rehabilitative tools were used previously if not using neuromodulation?

-Raw 116-118 “No study using other neuromodulation methods like transcranial alternating current stimulation (tACS), transcranial pulsed current stimulation (tPCS) or transcranial random noise stimulation (tRNS) were found in HD? Is this sentence needed? Did the authors include these methodologies in the search strategy? If not, then this information here is redundant.

-Raw 121 “RUL=right unilateral; BL=bilateral.” This information should be part of the abbreviations under the table.

-Table 1 is missing the proper referencing for studies, how can the reader find the reference if he/she only finds, for example “Evans et al. (1987)”? It should be Evans et al. (1987) [ref number]. Please revise the entire manuscript.

-Table 1 please provide full terms of acronyms under the table under abbreviation (such as TFC, NA, ECT, HAM-D, MMSE, mC, PANSS, BPRS..). There is some inconsistency in using full terms inside of the table, but it would be better to use acronyms consistently and to put all full terms under the table.

-Further, what does it mean “there is improvement in psychosis” in Evans et al study?

-dTMS is what in Table 1?

-Raw 294 is missing the ref number for Shukla et al. Also, for Brusa, the reference number is missing. For Grois, it should be put near the surname, not at the end of the sentence, the reference number. Also, raw 337 for Bocci, the ref number should be placed accordingly. Please check the entire manuscript.

-One of the missing issues for TMS studies is the location determination since there are accuracy differences if using TMS with EMG without navigation, line navigated TMS and e-field TMS. This needs to be elaborated on in the discussion.

-Raw 316, the authors need to insert newly updated references on the new stimulation protocols for TMS, such as theta burst. There is no explanation for the different stimulation protocols that exist there for TMS.

-Raw 356-357 , this sentence need to be written clearly “Current reports suggest that this intervention might play a role in the treatment of cognitive, motor and psychiatric symptoms of HD, especially depression and psychosis”. Since mainly psychiatric symptoms are assessed, the sentence does not need to include “cognitive” and “motor”.

-          Raw 353 “To summarize, due to the limited methodological quality of most studies, it is not 353

possible to draw definite conclusions about the effects of any particular neuromodulation 354

treatment for HD-related symptoms”. Please explain what do you mean by “quality”? This term is written several times throughout the manuscript.  

Author Response

Reviewer 3:

1-Abstract – it is really too stressful sentences such as “Despite eighteen out of nineteen studies showed improvement in HD symptoms, further investigations are required due to the quality of the available evidence and related methodological shortcomings. It is likely that HD-related symptoms can be treated with neuromodulation methods with little or controllable side effects.” Since authors are covering different psychological, cognitive and motor symptoms, it is difficult to determine what conclusions are related to what. The results can be presented as the efficacy of different studies in treating different symptoms. Which symptoms are more likely to be treated efficiently with neuromodulation techniques? What is the duration of the therapeutic effects are there any information on this? The reader really does not have a clear take-off message. Which symptoms are mostly studied and can be investigated therapeutically with what?

Response: We tried to reformulate the Abstract to convey a more specific message regarding the therapeutic use of neuromodulation techniques for HD (see line 25-30). 

2-In the introductory under the aims, the terms motor, cognition, and behavior-related symptoms are used. Later in the manuscript, the authors use “psychiatric symptoms”. It is suggested to introduce the proper terms and to be consistent with the presentation of the symptomatology versus testing outcomes (later presented in the Results section. In the introduction, the authors can introduce examples of symptomatology under superordinate terms such as “psychiatric symptoms,” or psychological symptoms, or behavioral symptoms, or motor symptoms.

Response: We agree that the use of a consistent terminology throughout the manuscript might help to reduce possible confusion to the reader. Therefore, we replaced the use of “psychiatric symptoms” with “behavioral symptoms” in the manuscript.

3-Under the searching strategy, the authors included TMS studies, but later on when presenting the authors can specify which TMS device was used: TMS with EMG without navigation, line navigate TMS or e-field navigated TMS?

Response: In the search strategy, we used the term “TMS” as it would yield search results including all modalities of TMS. In the new version of the manuscript, we specified the TMS protocols reported in the selected articles. Unfortunately, the studies did not include any TMS navigation modality. All studies reported target definition based on head surface anatomic landmarks. Regarding motor threshold, two studies reported the use of EMG but two other studies did not report the method used to determine the motor threshold. We tried to clarify this information in the new version of the manuscript (see lines 197-201).

4- What is the difference between rTMS and deep TMS? RTMS refers to stimulation protocol but deep refers to coil usage right? It is strange to use rTMS vs dTMS please check this. I believe H coil was related to author's explanation of dTMS, if yes this need to be corrected.

Response: Both terms refer to different aspects of the protocols used for intervention. rTMS refers to the repetition of pulses used in a course of TMS. Single pulse TMS uses a single monophasic magnetic pulse, whereas repetitive TMS, or, rTMS, refers to the application of repetitive biphasic magnetic pulses [Klomjai W et al. Ann Phys Rehabil Med 2015; 58(4): 208-13]. There is a wide array of rTMS protocols with different pulse frequencies and session duration. Conversely, dTMS often refers to deep repetitive TMS, that was first introduced by the use of H coils. These coils have the potential to reach deeper regions of brain tissue (i.e. aproximately 5.5 cm) [Bersani FS et al. Eur Psychiatry 2013; 28: 30-39.]. Because rTMS and dTMS are commonly used in the scientific literature and were also used in the selected studies, we employed these terms as originally reported. For a detailed description of the protocols used in each study, we also specified the type of coil and all other available parameters for each study protocol (i.e., pulse frequency, number of sessions, session duration, sham protocol) (see the 4th column -Intervention- of table 3 and lines 192-203 and 314-333).

5-Please state in the introductory part that none of the neuromodulation technique does not have FDA clearance used for treatment, the use of neuromodulating techniques in HD are still purely research-based.

Response: We addressed this issue in the current version of the manuscript (see lines 49-51).

6-Raw 105 “The most common reason for exclusion was not using neuromodulation as a treatment modality for HD”. Can authors explain which rehabilitative tools were used previously if not using neuromodulation?

Response: By this sentence, we meant that several studies used techniques, such as TMS, for neurophysiological research not therapeutics. We rephrased the sentence in the manuscript (see lines 111-112). As there are no disease modifying treatments for HD, management of HD is focused on amelioration of symptoms through pharmacological (e.g., vesicular monoamine transporter 2, VMAT2, inhibitors, and antipsychotics) and non-pharmacological approaches, included different psychosocial/educational interventions. We provide a brief summary of the strategies used in the management of HD in lines 40-48 of the manuscript.

7-Raw 116-118 “No study using other neuromodulation methods like transcranial alternating current stimulation (tACS), transcranial pulsed current stimulation (tPCS) or transcranial random noise stimulation (tRNS) were found in HD? Is this sentence needed? Did the authors include these methodologies in the search strategy? If not, then this information here is redundant.

Response: The search strategy included the umbrella term “electric stimulation” and other equivalent terms that would yield results from all kinds of electric stimulation. In this regard, this information is not redundant, had been requested by a reviewer in the previous submission, and should be kept in the manuscript.

8-Raw 121 “RUL=right unilateral; BL=bilateral.” This information should be part of the abbreviations under the table.

Response: The information regarding the abbreviations “RUL” and “BL” are provided in the table legend (see table 1 legend, line 128).

9-Table 1 is missing the proper referencing for studies, how can the reader find the reference if he/she only finds, for example “Evans et al. (1987)”? It should be Evans et al. (1987) [ref number]. Please revise the entire manuscript.

Response: We addressed this issue in the current version of the manuscript (see the 1st column – Author, date - of tables 1, 2, 3).

10-Table 1 please provide full terms of acronyms under the table under abbreviation (such as TFC, NA, ECT, HAM-D, MMSE, mC, PANSS, BPRS..). There is some inconsistency in using full terms inside of the table, but it would be better to use acronyms consistently and to put all full terms under the table.

Response: We addressed this issue in the current version of the manuscript (see table legends in lines 128-132).

11-Further, what does it mean “there is improvement in psychosis” in Evans et al study?

Response: The phrase was intended to highlight that the case reported by Evans et al. improved psychotic symptoms after ECT. We rephrased this sentence in the current version of the manuscript (see 6th column - Outcome – and first row of table 1).

12-dTMS is what in Table 1?

Response: The term dTMS refers to deep repetitive transcranial magnetic stimulation (see lines 329-331 in the new version of the manuscript).

13-Raw 294 is missing the ref number for Shukla et al. Also, for Brusa, the reference number is missing. For Grois, it should be put near the surname, not at the end of the sentence, the reference number. Also, raw 337 for Bocci, the ref number should be placed accordingly. Please check the entire manuscript.

Response: We addressed these issues in the current version of the manuscript (see lines 314-333 and 357-359).

14-One of the missing issues for TMS studies is the location determination since there are accuracy differences if using TMS with EMG without navigation, line navigated TMS and e-field TMS. This needs to be elaborated on in the discussion.

Response: As requested, we addressed the issue of stimulus accuracy in TMS as one of the main limitations of TMS in the Discussion section (see lines 319-321). In the new version of the manuscript, we also mentioned that all studies used anatomic landmarks for target definition (See lines 197-201).

15-Raw 316, the authors need to insert newly updated references on the new stimulation protocols for TMS, such as theta burst. There is no explanation for the different stimulation protocols that exist there for TMS.

Response: We understand the reviewer’s concern and tried to better discuss this topic in the new version of the manuscript (see lines 334-340).

16-Raw 356-357 , this sentence need to be written clearly “Current reports suggest that this intervention might play a role in the treatment of cognitive, motor and psychiatric symptoms of HD, especially depression and psychosis”. Since mainly psychiatric symptoms are assessed, the sentence does not need to include “cognitive” and “motor”.

Response: We rephrased this sentence in the current version of the manuscript (see lines 382-387).

17-          Raw 353 “To summarize, due to the limited methodological quality of most studies, it is not 353 possible to draw definite conclusions about the effects of any particular neuromodulation 354

treatment for HD-related symptoms”. Please explain what do you mean by “quality”? This term is written several times throughout the manuscript.

Response: The term quality refers to the quality of the studies. We used Joanna Briggs Institute’s (JBI) critical appraisal tools to access the quality of all studies included in the review (see methods, lines 99-106, and see supplemental material for the complete table of the quality assessment). The instrument rates the quality of the studies based on multiple parameters that are related to design and methodological aspects of the studies.  

Round 2

Reviewer 1 Report (Previous Reviewer 2)

I think this manuscript  would be suitable for publication.

Reviewer 3 Report (New Reviewer)

The authors cleared the raised comments.

This manuscript is a resubmission of an earlier submission. The following is a list of the peer review reports and author responses from that submission.

Round 1

Reviewer 1 Report

This study conducted a systematic review of the effect of non-invasive neuromodulation methods on the cognitive, motor, and behavioral symptoms of Huntington’s disease. The authors showed that non-invasive neuromodulation methods have a positive trend for symptoms and presented important data that suggest the effectiveness of neuromodulation methods as a treatment for HD. I am interested in this research, but I don’t recommend to proceed next step. The reason is there are few previous studies, and these have big heterogeneity (as described in Lines 188-189). So it is not sufficient to conduct a systematic review of this field. More than half of the studies were case studies, and when these subdivide by the type of NIBS, stimulation sites, or outcome, it is not a high enough number to conduct a systematic review. Therefore, I think it is too early to perform a systematic review in this field. Even when looking at the Discussion, the authors seem not to discuss enough the results of this study. I have doubts about what significance of the clinical field this study has and how this study can advance knowledge to date.

The Introduction and Discussion are not sufficient and not constructive. The concept of non-invasive neuromodulation is to modulate abnormal cortical activation to a normal state. So, the authors should review the abnormal brain activity observed in HD in the Introduction. Based on that, in the Discussion, the authors should describe whether the results of the NIBS effect obtained from this study were valid regarding abnormal cortical activity observed in HD. The NIBS can modulate cortical activity bidirectionally. It is necessary a brief explanation of bidirectional modulation by NIBS.

Reviewer 2 Report

This report summarized the evidence on the potential benefits of brain stimulation on clinical symptoms in patients with Huntington's disease.

1) Please describe the inclusion and exclusion criteria more in detail.

2) Please describe the electrophysiology of Huntington's disease, and potential benefits of brain stimulation on this etiology. For example, patients with depression sometimes present hypoactivity in dorsolateral prefrontal cortex, and high frequency repetitive transcranial magnetic stimulation over dorsolateral prefrontal cortex enhances cortical excitability in this area, which contributes to the improvement in depressive symptoms.

3) Please describe what kind of ECT protocol is optimal to improve some symptoms in this disease. Some previous reports showed that ECT improves depressive symptoms and suicide ideation in patients with HD, but I think those symptoms were not always typical in patients with HD. Please explain whether these symptoms were due to organic factors due to HD, depression, or normal psychological reaction after the patients were suffering from HD. Furthermore, please describe whether ECT was safe in the motor symptoms in those patients, because some reports explained that ECT worsened motor symptoms.

4) Please explain what kind of rTMS protocol was optimal in the treatment of HD. First, high frequency rTMS usually enhances cortical excitability while low frequency rTMS inhibits it. Then, which is better to improve HD and why? 

5) I think previous studies may not be optimized in views of rTMS protocol. What should the motor threshold be and how many TMS sessions were optimal in improving specific symptoms in HD? 

6) In the ECT, rTMS, and tDCS please describe more clearly what kind of protocol improved some symptoms and what kind of other protocol did not improve it. Please describe more concisely what kind of studies are warranted. What kind of protocol in future studies is probably best to improve symptoms in those patients? (This protocol should include the name of device, the frequency, motor threshold, intensity, number of sessions, localization and so on).

I think it is necessary to revise the manuscript.